# Fermented cereal-based Munkoyo beverage: Processing practices, microbial diversity and aroma compounds

Sydney Phiri[1,2], Sijmen E. Schoustra[1,2]*, Joost van den Heuvel[1], Eddy J. Smid[1], John Shindano[2], Anita Linnemann[1]

**1** Wageningen University and Research, Laboratories of Genetics, Food Microbiology and Food Quality and Design, Wageningen, The Netherlands, **2** Department of Food Science and Nutrition, School of Agricultural Sciences, University of Zambia, Lusaka, Zambia

* Sijmen.Schoustra@wur.nl

**Data Availability Statement:** All the raw sequence data are deposited at the NCBI SRA under bioproject number PRJNA575741; http://www.ncbi.nlm.nih.gov/bioproject/575741.

## Abstract

Fermented cereal-based foods play a crucial role in attaining food and nutrition security for resource-poor populations in sub-Saharan Africa. These products are widely produced by spontaneous fermentation using of cereal grains as raw material. They have a unique taste and flavour, are rich sources of energy and their non-alcoholic nature makes them ideal for consumption by the entire population, including children. Lactic acid bacteria dominate the fermentation process and lead to a low pH of around 4, which suppresses the growth of pathogenic bacteria, thereby increasing the shelf-life and safety of the food. Knowledge about processing practices, consumption patterns and bacterial communities is essential to regulate processing and design appropriate mixes of micro-organisms to produce starter cultures for commercial production of standard-quality fermented foods that meet desired quality characteristics. In four regions of Zambia, we surveyed processing practices and consumption patterns of a spontaneously fermented cereal-based beverage called Munkoyo, commonly produced in Zambia and the Democratic Republic of Congo. Variations in processing practices exist in cooking time of the unfermented maize porridge and time allowed for fermentation. Consumption is mainly at household level and the product is considered as an energy drink. Characterisation of the bacterial communities of over 90 samples with 16S amplicon sequencing on DNA extracted from the entire bacterial community revealed six dominant families, namely *Streptococcaceae*, *Leuconostocaceae*, *Enterobacteriaceae*, *Lactabacillales*, *Bacillaceae* and *Aeromonadaceae*, and a Shannon index of up to 1.18 with an effective number of 3.44 bacterial species. Bacterial communities that underlie the fermentation in Munkoyo differ in their composition for the different regions using common processing steps, suggesting that different combinations of bacteria can be used to achieve successful Munkoyo fermentation. Analysis of aroma profiles in 15 different samples from two different Provinces showed that aldehydes, esters, organic acids, alkanes, alkenes and alcohols dominated.

**Funding:** Funding was provided by NWO-WOTRO (Netherlands Organization for Scientific Research Science for Global Development Division), grant number WOTRO 08.250.2013.108 to Sijmen E. Schoustra.

**Competing interests:** The authors have declared that no competing interests exist.

## Introduction

The production of fermented beverages is one way to utilize cereals for human consumption. Fermentation is known to restrict the proliferation of bacterial pathogens, resulting in an increased shelf-life and microbial safety of these products. The main mechanism for this functionality is that of lowering the pH to values below 4 by the production of lactic acid bacteria [1]. Moreover, fermentation leads to a generally perceived improvement in texture, taste and aroma of the final product due to the development of a complex blend of texture and flavour compounds [1, 2]. In addition, advances in scientific knowledge have taught us other benefits of the activities of micro-organisms in food preparation, such as the development of health supporting properties due to vitamin production and antidiarrheal attributes [3].

Around the world, many traditional cereal-based fermented beverages exist, both alcoholic and non-alcoholic [4–6]. These beverages are frequently consumed because they are inexpensive to prepare and do not require refrigeration or pre-heating prior to consumption [6]. As a result of the appetizing taste and flavour, adults and, in the case of the non-alcoholic products, children alike consume cereal-based beverages, for instance at major ceremonies such as weddings or funerals. The fact that the preparation of these beverages is based on spontaneous fermentation entails that the process is not controlled regardless of the vessel used. This leads to a diverse microbial flora from the local environment, besides variations in the production process. The way in which microbial communities develop during spontaneous fermentation, depends on the food ingredients and the surrounding environment in addition to the interaction of the micro-organisms themselves [4, 6, 7]. In non-alcoholic products, lactic acid bacteria dominate, of which the predominant lactic acid bacteria (LAB) in most cereal-based fermented beverages include *Lactobacillus*, *Lactococcus*, *Leuconostoc* and *Pediococcus*. The *Lactobacillus* species include *L. fermentum*, *L. plantarum* and *L. delbrueckii* [2, 6].

Sensorial properties of fermented foods are largely determined by micro-organisms, which makes identification of the microbial communities and their diversity crucial to understand how flavour and taste of these products come about. Aroma compounds are a key component of these sensorial properties [2, 4]. These aroma compounds consist of many volatile and non-volatile compounds, which possess diverse chemical and physicochemical properties. They include alcohols, aldehydes, esters, di-carbonyls, short to medium-chain free fatty acids, methyl ketones, lactones, phenolic and sulphur compounds. The non-volatile compounds largely contribute to the taste, whilst volatile compounds influence both the taste and flavour [8–10]. Especially the presence of diacetyl, acetic acid and butyric acid makes fermented beverages appetizing to a large group of consumers [9].

Most African traditional fermented beverages are widely consumed and embedded in the local culture [2]. They are mostly produced at household level. Details on processing practices and/or fermenting microbes are largely unreported, although some have been documented [4, 7, 11–13]. As a result, little is known about the composition and diversity of micro-organisms that drive the fermentation process. Lack of this information impedes formal and up scaled development of these products, resulting in, amongst others, a rapidly increasing number of mostly urban consumers who do not have access to the traditional products that are part of their cultural heritage. Rapid urbanisation leads to an increasing urgency to take the production of traditional and culturally embedded foods to the next level.

This is also true for Munkoyo, a traditional cereal-based fermented beverage from Zambia and the Democratic Republic of Congo [14, 15]. Previous work has described the general features of processing and has found that various groups of LAB such as *Lactobacillus plantarum*, *Weissella confusa*, *Lactococcus lactis* and *Enterococcus italicus* are present in the final product [15]. These microbes likely drive the fermentation and hence determine the type of final

product and its properties. The first aim of this paper is to survey the current state-of-the-art in the production of Munkoyo from different regions in Zambia and to establish the main reasons for consumption. The second aim is to measure physiochemical properties and to profile Munkoyo samples for their bacterial community composition, as linked to sampling location, variations in processing practices and consumption preferences. The third aim is to identify the aroma compounds of the beverage that provide the unique sensory attributes to Munkoyo. Knowledge about microbial communities and their processing practices is essential to be able to standardize production of the beverage and advances the understanding of the factors that drive the species composition of fermenting microbes. A standardized production process will benefit from the use of defined mixes of micro-organisms (i.e. starter cultures), which facilitate controlled production of fermented foods to optimally meet the sensory quality characteristics desired by consumers [5].

## Materials and methods

### Questionnaires on traditional processing practices

A questionnaire was designed (S1 Questionnaire) to assess prevailing processing procedures as well as consumption preferences and patterns of Munkoyo. Questionnaires were administered to consumers and processors of Munkoyo at 4 locations in Zambia: Mumbwa, Chibombo, Lusaka and Chongwe. All locations are in the same middle rainfall agro-ecological zone (II) (Fig 1). Respondents were selected by the 'snowballing' method, defined as a nonprobability sampling technique that cannot be statistically signified [16]. Camp Agricultural Extension officers (CEOs) selected participants based on their willingness to participate in the research. A total of 172 participants aged between 19 and 60 years, comprising 62 males and 110 females,

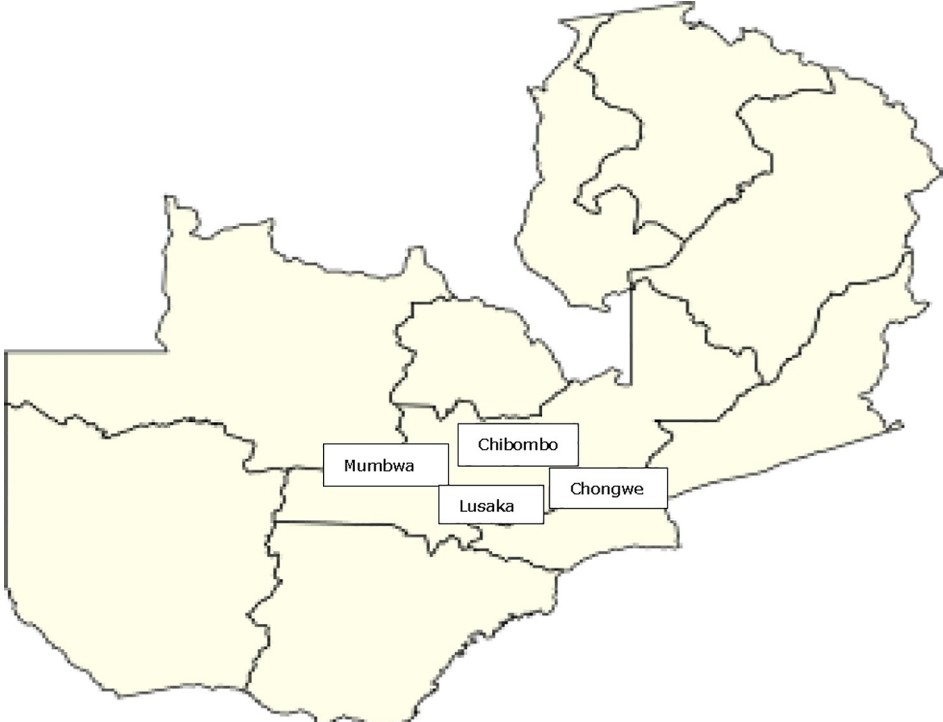

**Fig 1. Sampling locations.** Middle rainfall agro-ecological zone(II) where questionnaires were administered and samples of Munkoyo collected to assess the prevailing processing procedures and consumption pattern.

filled in the questionnaires, which included questions on raw materials, treatment steps, timing of these steps, type of fermentation vessel and the addition of other materials to facilitate fermentation, such as roots from a special plant called *Rhynchosia*. Questions on consumption patterns and/or use of Munkoyo focussed on consumed quantities, who mostly consumed it, at what occasions and main motivation for consumption. Ethical clearance for this work was granted by the Directorate for Research and Graduate Studies, University of Zambia, dated 31 January 2017.

## Physical properties and analysis of the microbial species

Munkoyo samples were collected from more than 50% of the questionnaire respondents randomly selected from the four locations of the research, amounting to 96 samples for analysis of pH, titratable acidity and bacterial species composition. pH was recorded using a portable pH meter (*Mettler Toledo AG*). Titratable acidity was determined by titrating 10 cm$^3$ of the sample against sodium hydroxide with phenolphthalein as an indicator. Bacterial species composition was determined based on 16S DNA amplicon sequencing. For this, all DNA of the bacteria was extracted from each sample, following Schoustra et al. [15] as follows. One ml of Munkoyo sample was spun down at high speed, after which the supernatant was discarded. Next, 500 μl TESL, 10 μl mutanolysin solution and 100 μl lysozyme solution were added to the pellet and incubated at 37˚C for 60 min with slight shaking. Then 500 μl GES reagent was added, followed by cooling on ice for 5 min. Subsequently 250 μl of cold ammonium acetate solution was mixed gently, followed by keeping the mixture on ice for 10 min before spinning and collecting the supernatant. The supernatant was purified with chloroform-2-pentanol by mixing 1:1, spinning down at 12,000 rpm and collecting the supernatant. DNA was precipitated by adding 0.1 volume of 3 M sodium acetate followed by 2.5 volumes of 100% ethanol, and storage at -20˚C overnight. Next the mixture was spun for 20 min at 12,000 rpm at 4˚C and the supernatant removed. Finally, the DNA pellet was washed out by adding 1 ml of cold 70% ethanol and spinning for 10 min at 12,000 rpm at 4˚C. After removal of the supernatant, the DNA pellet was air dried for 10 min at room temperature, dissolved in 10 mM Tris pH 7.5. The purified DNA was sent to LGC Genomics in Berlin, who performed 16S amplicon sequencing. They first performed a PCR using about 1–10 ng of DNA extract (total volume 1 μl), 15 pmol of each forward primer (341F; `CCTACGGGNGGCWGCAG`) and reverse primer (785R; `GACTA CHVGGGTATCTAATCC`) (in 20 μL volume of 1 x MyTaq buffer containing 1.5 units MyTaq DNA polymerase (Bioline) and 2 μl of BioStabII PCR Enhancer (Sigma)). For each sample, the forward and reverse primers contained a unique 10-nt (company specific) barcode sequence. PCRs were carried out for 30 cycles, using the following parameters: 2 min 96˚C pre-denaturation; 96˚C for 15 s, 50˚C for 30 s, 70˚C for 90 s. About 20 ng amplicon DNA of each sample were pooled for up to 48 samples carrying different barcodes. If needed, PCRs showing low yields were further amplified for 5 cycles. The amplicon pools were purified with one volume AMPure XP beads (Agencourt) to remove primer dimer and other small mispriming products, followed by an additional purification on mini elute columns (Qiagen). About 100 ng of each purified amplicon pool DNA was used to construct Illumina libraries using the Ovation Rapid DR Multiplex System 1–96 (NuGEN). Illumina libraries were pooled and size selected by preparative gel electrophoresis. Sequencing was done on an Illumina MiSeq using V3 Chemistry (Illumina). Raw data were obtained from LGC Genomics and served as input for data analysis, for which we developed a pipeline. Bioinformatics and data analysis generated by DNA sequencing went through a rigorous quality system, which involved identification and removal of sequences containing more than one ambiguous base (N) and evaluation of the presence and complementarity of primer and barcode sequences. For further data processing and

statistics the QIIME pipeline [17], modified from Bik et al [18] was used. Paired-end reads were joined using join_paired_ends.py (with minimum overlap 10 basepairs) after which sequences were trimmed and filtered using cutadapt (v1.11 -q 20, -m 400, Martin 2011) using the known primer sequences `CCTACGGGNGGCWGCAG` and `GACTACHVGGGTATCTAAKCC` to trimmed both sides of the sequence. These trimmed sequences were then checked for chimera's, using uchime (v4.2.20, gold database)[19], with sequences with a lower chimera score than 0.28 were retained. After these trimming and filtering steps sequences were clustered into operational taxonomic units (OTUs) after quality check using pick_open_reference_otus.py (-s 0.1, -enable_rev_strand_match TRUE, -align_seqs_min_length 75, -pick_OTU_similarity 0.95). Taxonomy of the resulting OTUs was assigned to representative sequences using the Greengenes (v13.5) rRNA database. This algorithm gives a representative sequence for an OTU, which were used to perform a local blast using the gold database from uchime. The taxonomy from the top BLAST hit was used for further data processing. The anosim nonparametric test from the vegan package, use from the compare_categories.py wrapper in QIIME was used to test for significant differences in OTU tables between treatment factors (1000 permutations). Furthermore we used group_significance.py to perform a Chi-square test for differences in abundance between individual OTUs between treatment levels.

## Analysis of aroma compounds

Fifteen samples of Munkoyo, representative of all four surveyed locations, were selected for aroma analysis. Samples were defrosted and put in triplicate in 2 ml GC-MS vials, which were subsequently tightly closed to avoid loss of aroma compounds. Next the volatile compounds were extracted for 20 min at 60°C using a SPME fibre (Car/DVB/PDMS, Supelco). The compounds were desorbed from the fibre for 2 min on a Stabilwax- DA-Crossbond-Carbowax-polyethylene-glycol column (30 m length, 0.25 mmID, 0.5 μm df). The gas chromatograph settings were: PTV Split-less mode (5 min) at 250°C. The carrier gas was helium with a constant flow of 1.5 ml/min. The GC oven temperature was set at 40°C for 2 min and later raised to 240°C (10°C/min) and kept at this temperature for 5 min. Mass spectral data was collected over a range of m/z 33–250 in full scan mode with 3.0030 scans/sec. GC-MS results were analysed using Chromeleon 7.2 software. The generated signal peaks were identified as aromatic products according to their elution time and relative area peaks. Using a library in Chromeleon, the names of the aroma compounds were identified. Area peaks, molecular weights and retention times were recorded.

## Statistical analysis

The Statistical Package for Social Sciences (IBM SPSS statistic 23) and Microsoft Excel 2016 were used to analyse the responses from the questionnaires. Chi-square test generated the degrees of freedom (df) and the p-value determined the significant differences in the processing methods and consumption patterns.

## Results

### Processing practices and consumption patterns

This study systematically assessed the processing of Munkoyo using questionnaires of 172 consumers and producers at four locations in Zambia. Common steps were found to include soaking of maize meal in water, cooking of the mixture, cooling and temporary incubation with *Rhynchosia* roots as shown in Fig 2.

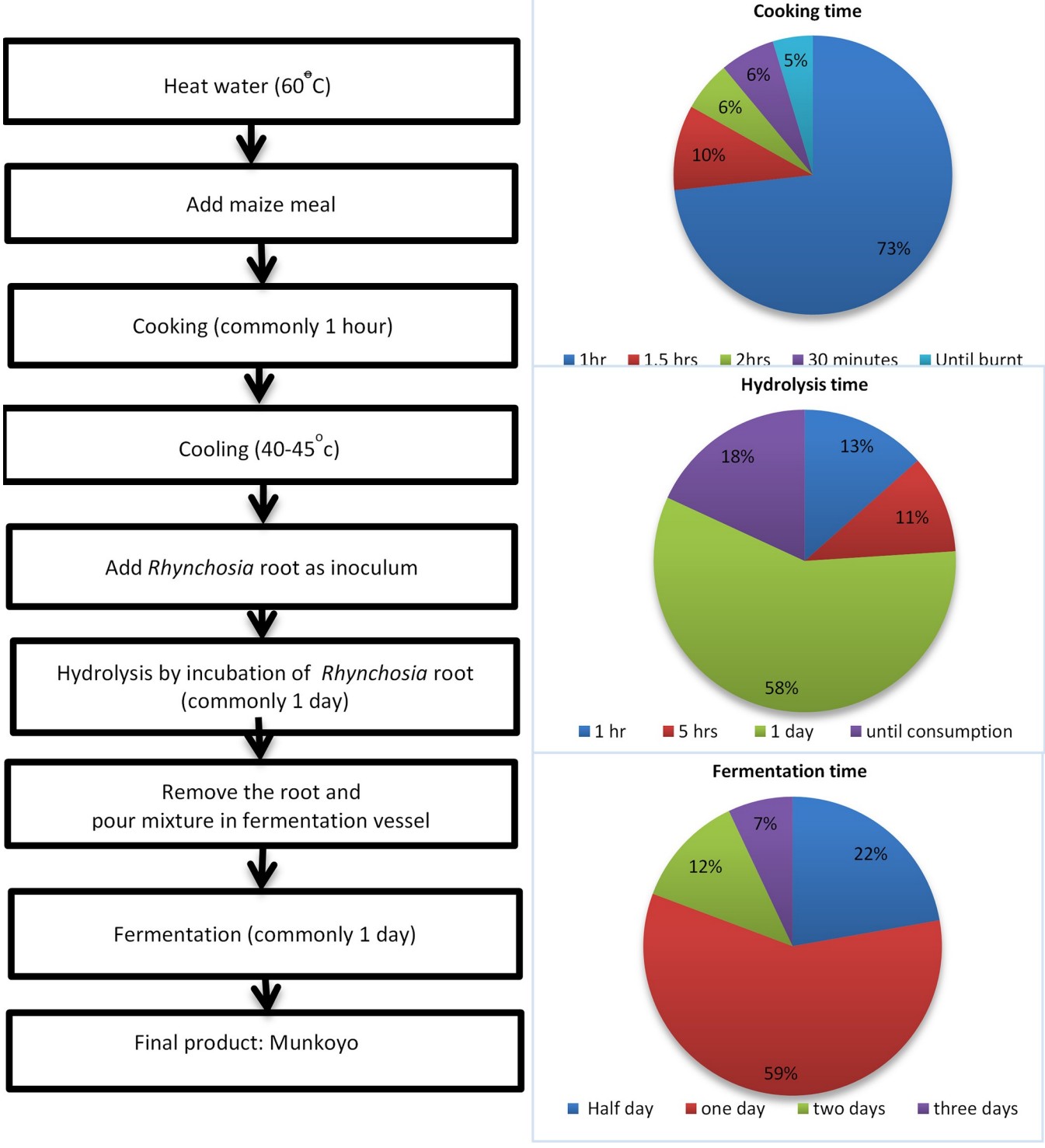

**Fig 2. Flow diagram.** Processing steps of making Munkoyo and pie charts showing common variations in the duration of cooking the maize/water mixture, hydrolysis by the enzymes from *Rhynchosia* roots and fermentation in the fermentation vessels (buckets or calabashes).

The main variations in processing concern the duration of three critical steps: cooking of the initial maize porridge (cooking time), incubation with *Rhynchosia* roots (hydrolysis time)

and fermentation by micro-organisms (fermentation time). The cooking time ranged from 30 min to several hours; most common duration was one hour. This cooking time is known to be adequate to gelatinize the starch for the action of enzymes supplied by *Rhynchosia* roots [20]. Roots were beaten and stripped off prior to addition to the gelatinized porridge to increase the surface area of the roots, allowing the release of amylolytic enzymes into the water/maize mix to degrade the gelatinized starch into fermentable sugars. A temperature of around 45˚C is optimal for hydrolysis of starch. Time allowed for incubation with roots varied between one hour and several days, most commonly and practically done overnight. *Rhynchosia* roots were not the only sources used to supply amylolytic enzymes; around 5% of surveyed producers used millet malt, cowpea roots and/or sweet potato peels. Finally, the mixture with fermentable sugars was incubated in fermentation vessels for spontaneous fermentation for a period ranging from half a day to two days (most commonly one day) before consumption, and ongoing for mostly three days until the beverage was consumed completely.

The most commonly used fermentation vessel was the plastic container as indicated in Fig 3, because it is easily available. However historically, mostly calabashes were used. Calabashes are still used on occasion, especially by producers aiming at sales, because previously used calabashes are known to quickly ferment the beverage as they harbour bacteria on the walls of the calabash. Earthen ware and metal buckets are no longer common as fermentation vessel.

Fig 4 shows consumption patterns in different regions. The beverage is generally considered as an energy drink or snack, but is also consumed as a special drink at social gatherings such as weddings and funerals. However, people in formal employment with alternative options would only consume Munkoyo beverage at such social gatherings. The beverage is consumed by men and women of all ages, including children. They consume Munkoyo mainly when feeling hungry and at household level, which includes when working on the fields or during long distance travelling.

The observed variations in processing in relation to the time of cooking, hydrolysis, fermentation and variations in consumption patterns per sampling location or province were statistically analysed as shown in Table 1.

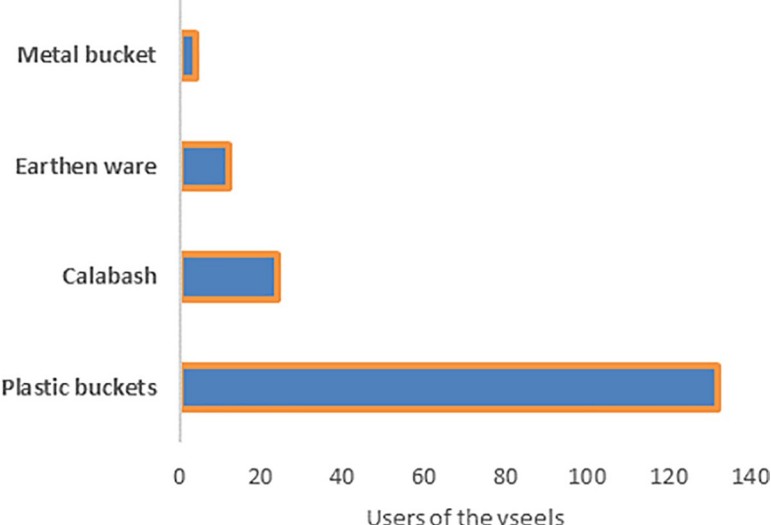

**Fig 3. Fermentation vessels.** Frequency of the type of fermentation vessel used for the production of Munkoyo, as number of users among the respondents of the questionnaire. Plastic buckets, which are easily available and durable, are the most commonly used vessels as compared to calabash, earthen ware or metal pot.

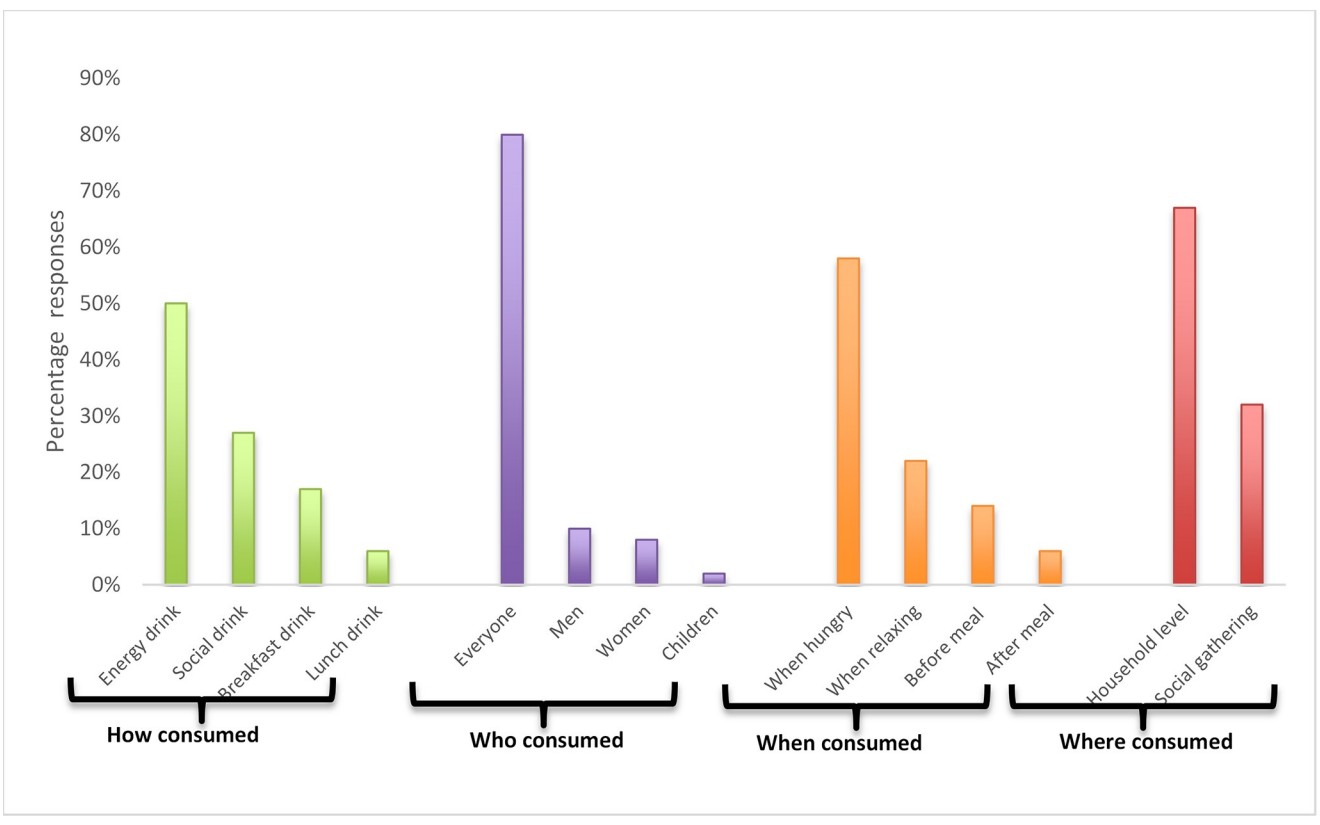

**Fig 4. Consumption patterns.** Consumption of Munkoyo in the four surveyed locations outlining how the beverage was consumed, who consumed it, and when and where it was consumed.

We found no significant differences between sampling locations in terms of cooking time, fermentation time, how the beverage is consumed, who consumed it and where it is consumed. However, there was a significant difference in different provinces and locations with respect to the hydrolysis time and when the beverage was consumed. The difference in hydrolysis time could be as a result of using another type of inoculum than *Rhynchosia* roots (i.e. millet malt, cowpea roots, sweet potato peels) for hydrolysis. The difference in when the beverage is consumed could be due to a difference in the social status of the consumers between locations and provinces. High social status groups tend to consume Munkoyo when relaxing whilst the majority low status groups consume Munkoyo as an energy giving food when they are hungry.

## pH and titratable acidity

The pH of Munkoyo samples from different locations ranged between 3 and 4 and titratable acidity between 0.2 and 1% (Fig 5). For both parameters, the values are statistically different per sampling location (ANOVA: pH, $F_{3,44}$ = 7.31, P = 0.00044; TTA, $F_{3,44}$ = 12.61, P < 0.0001). These pH and titratable acidity ranges have been recorded for most cereal-based beverages in Africa, such as Mangisi of Zimbabwe with a pH of 3.98, titratable acidity of 0.67% and lactic acid concentration of 4.10 g/L [21].

## Microbial composition and diversity

The bacterial composition of 96 samples was assessed based on 16S DNA amplicon sequencing of the V3-V4 region of the gene coding for the 16S RNA ribosomal subunit. This procedure

**Table 1. Statistical analysis.** Chi-square test indicating the relationship between processing parameters and consumption patterns of Munkoyo. Processing parameters and consumption patterns with P-values in the same row with different letters are significantly different (p-value<0.05; α after Bonferroni correction to correct for multiple testing is 0.05/14 = 0.0035).

| Contrast | Statistics | Cooking time | Fermentation time | Hydrolysis time | How Consumed | Who Consumed | Where Consumed | When Consumed |
|---|---|---|---|---|---|---|---|---|
| **Province** | Chi-square | 1,737 | 4,667 | 13,542 | 10,139 | 5,838 | 0.107 | 19,879 |
| | df | 3 | 2 | 2 | 3 | 3 | 1 | 3 |
| | p-value | 0.629[b] | 0.097[b] | 0.001[a] | 0.067[b] | 0.120[b] | 0.743[b] | 0.000[a] |
| **Location** | Chi-square | 2,717 | 5,343 | 7,222 | 6,757 | 6,912 | 0.527 | 28,468 |
| | df | 3 | 2 | 2 | 2 | 3 | 1 | 3 |
| | p-value | 0.437[b] | 0.069[b] | 0.027[a] | 0.080[b] | 0.075[b] | 0.468[b] | 0.000[a] |

yields over 10,000 reads per sample of around 350 base pairs each. Using bioinformatics analysis, we counted the number of unique sequence types (i.e. operational taxonomic units or OTUs) within each sample. We blasted the DNA sequence of each obtained read to a database to determine which species this unique sequence type had the highest similarity. Different unique types (OTUs) that blasted to the same species family were taken together when estimating diversity (the number of species present and their relative distribution). The analysis revealed over 42 different bacterial species within the microbial communities of the samples. In general, each sample contained up to eight dominant types, which each occupied at least 2% of the total population (Fig 6).

In several cases, species were represented by more than one type (i.e. OTU blasting to the same species), so at the level of unique types, the diversity was higher. Diversity measures include Shannon index (H), evenness and effective number of species (ENS). These measures were similar between sampling locations, highlighting that only slight variations in species distribution existed between the sampling locations. These differences between sampling locations were not statistically significant as observed in Table 2. (One-Way ANOVAs; Shannon $F_{3,69} = 1.751$, P = 0.147; Evenness $F_{3,70} = 0.724$, P = 0.541; ENS $F_{3,70} = 1.846$, P = 0.147). Additionally, the Shannon index number is of importance in projecting total biodiversity when it is used to estimate the effective number of species [22]. This estimated Effective Number of Species (ENS) is calculated by getting the exponential of the Shannon index, mathematically expressed as: $ENS = e^H$ [23].

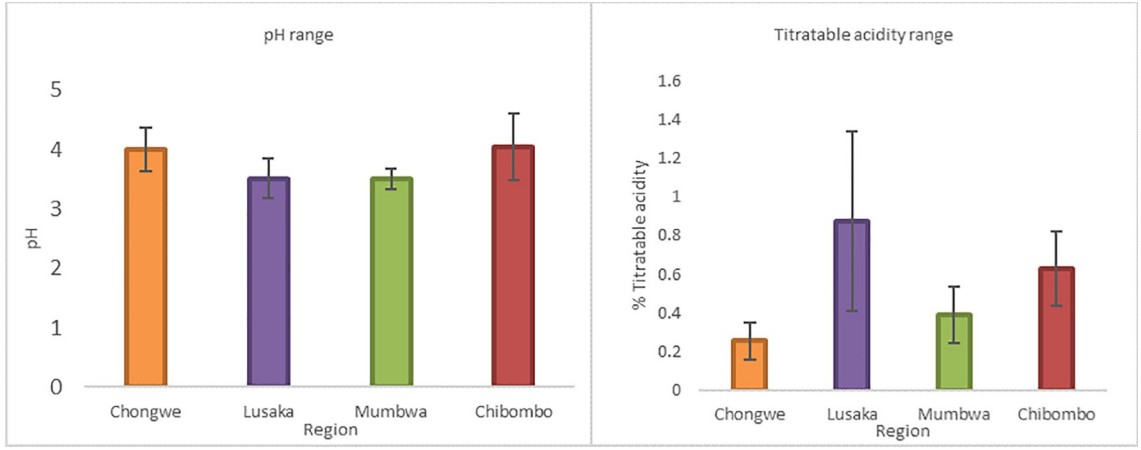

**Fig 5. pH values and titratable acidity (TTA) of Munkoyo samples from Chongwe, Mumbwa, Chibombo and Lusaka.** Error bars show standard deviation. The values for pH and for TTA are significantly different per sampling location.

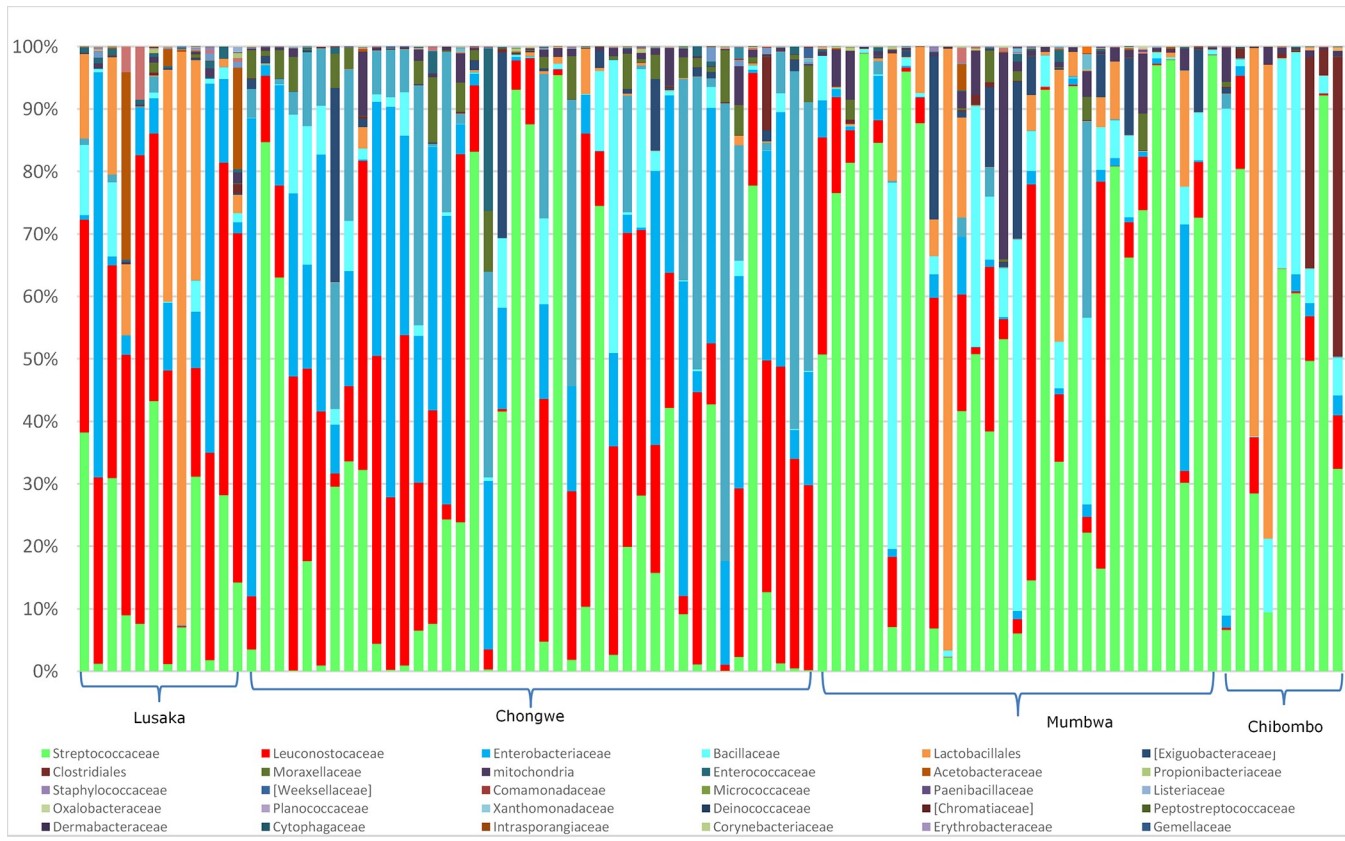

**Fig 6. Bacterial composition in Munkoyo from different regions.** Bars show relative abundance. Different colors show different families. Within each family multiple OTUs as well multiple species exit. *Streptococcaceae* were present in all the samples. *Leuconostocaceae* and *Enterobacteriaceae* dominated in samples from Lusaka province (Lusaka and Chongwe) whilst *Lactobacillales* and *Bacillaceae* dominated in samples from Central province (Mumbwa and Chibombo).

Thus, ENS estimates real biodiversity. In this case an ENS of 3.44 and 3.11 for Lusaka and Chongwe, respectively, indicates more diversity compared to Chibombo and Mumbwa with an ENS of 2.47 and 2.57, respectively. A Venn diagram (Fig 7) indicates at least 21 common species between the two provinces, with Lusaka province having more uncommon species across the regions (24 and 13 species) than Central province (14 and 4 species).

We performed an anosim to determine whether the variation observed in the microbial communities among the samples can be attributed to sampling location and the processing and consumption variables.

Results in Table 3 show that clustering both sampling location and when the product is consumed explain a significant part of the variation in the observed bacterial community structure. It should be noted, however, that when the product is consumed also varied significantly

**Table 2. Diversity indices.** Average Shannon index (H), evenness and effective number of species (ENS) of all samples per sampling location. Numbers between brackets show standard deviation.

| Location | Shannon index (H) | Evenness | ENS |
|---|---|---|---|
| Chibombo | 0.86 (0.30) | 0.27 (0.23) | 2.47 (0.71) |
| Lusaka | 1.18 (0.33) | 0.28 (0.080) | 3.44 (0.98) |
| Chongwe | 1.14 (0.45) | 0.43 (0.64) | 3.11 (1.43) |
| Mumbwa | 0.87 (0.52) | 0.21 (0.12) | 2.57 (1.38) |

**Fig 7. Species abundance comparison between provinces.** Venn diagrams showing overall intersection of bacterial species between Lusaka and Central province with a different number of uncommon species in each of the provinces.

by location, resulting in an autocorrelation in this analysis. In contrast, processing variables do not explain the variation in observed microbial community structure, nor do parameters related to the use of the product.

## Aroma compounds in Munkoyo

To further characterize Munkoyo and its variations, aroma profiles of 15 representative samples from all surveyed locations with different commonly used inoculum treatments were measured. Full profiles of aroma compounds are in Fig 8. The most abundant compounds include hexane, ethyl acetate, 2–3 butanedione, acetic acid, 2, 3 pentanedione, ethanol, hexanal and 2-n-pentylfuran. The produced aroma compounds were aldehydes, esters, organic acids, alkanes, alkenes and alcohols as indicated in Table 4.

## Discussion

In this study we reported the results of a survey documenting the production practices and consumption patterns of Munkoyo from different regions in Zambia; assessed physicochemical properties and profiled the bacterial communities and the aroma compounds in Munkoyo, which linked variations in processing and consumption preferences to variations in bacterial communities involved in the spontaneous fermentation process; and determined the aroma profile of a subset of samples.

The main variations in processing practices include time for cooking of the maize/water mix (cooking time), time for incubation of enzymes from *Rhynchosia* roots to allow

**Table 3. Hierarchical cluster analysis for impact of treatment variables such as location, processing and consumption on OTU tables.** Variable, test statistic (R), degrees of freedom (Dfs) and exact p value (P) are given, unless the p value was smaller than 0.001, which is indicated by <0.001.

| Variable | R | Dfs | P |
|---|---|---|---|
| Sampling location | 0.208 | 4,87 | <0.001 |
| Province | 0.293 | 2,87 | <0.001 |
| Cooking time | 0.059 | 5,87 | 0.106 |
| Hydrolysis time | 0.072 | 4,87 | 0.051 |
| Fermentation time | 0.015 | 3,87 | 0.32 |
| How consumed | 0.035 | 4,87 | 0.107 |
| Who consumed | 0.035 | 4,87 | 0.241 |
| Where consumed | -0.023 | 2,87 | 0.741 |
| When consumed | 0.129 | 4,87 | <0.001 |

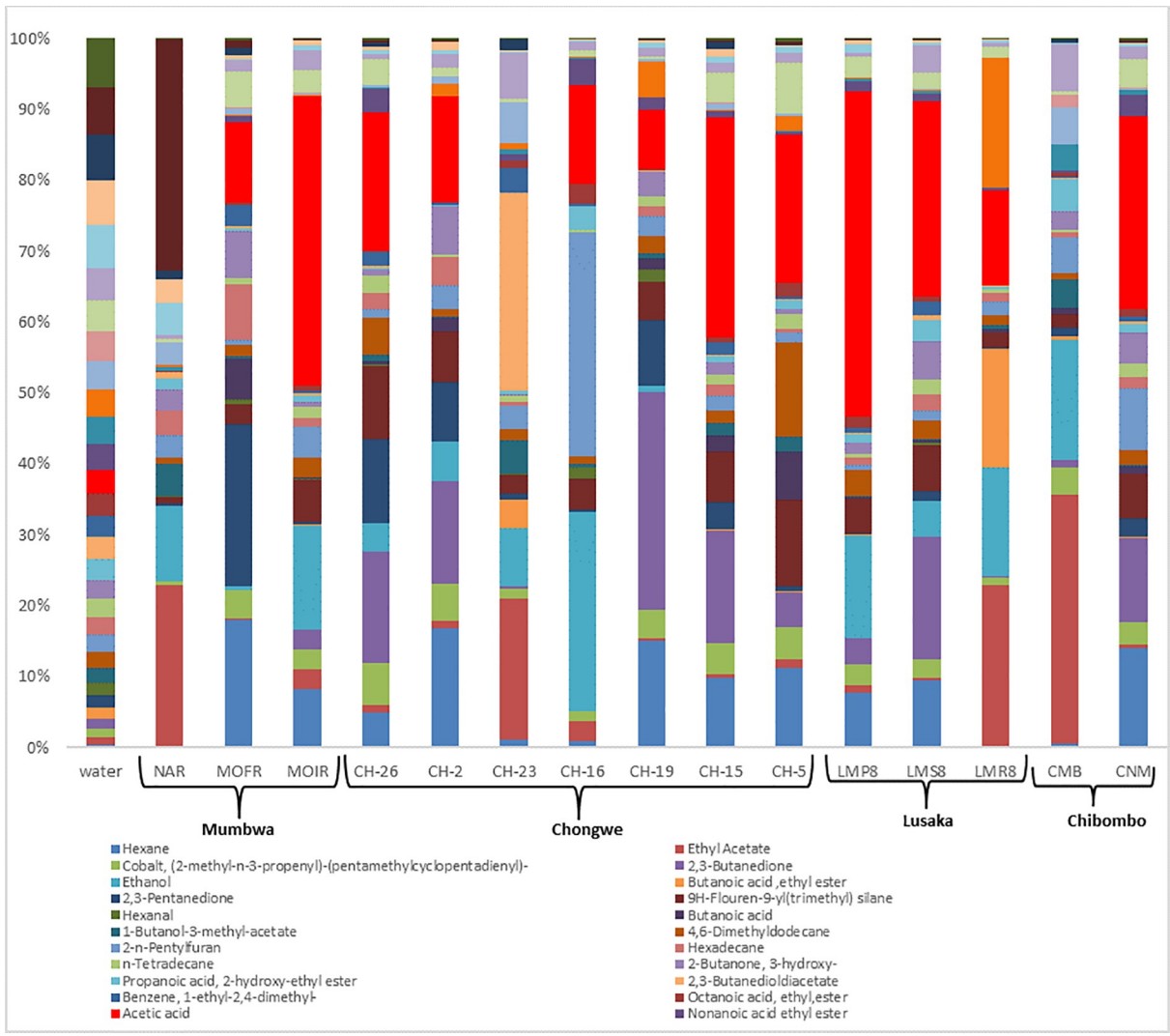

**Fig 8. Aroma compounds in Munkoyo.** Results show acetic acid and 2,3-butanedione dominating in the samples.

degradation of gelatinized starch (hydrolysis time) and time allowed for fermentation by bacteria (fermentation time). The pH of Munkoyo ranged from 3.8 to 4.2 and TTA ranged from 0.2% to 0.8%. This low pH and high acidity are known to reduce the proliferation of most pathogenic bacteria [24, 25]. This may explain why there are seldom cases of pathogenic contamination in Munkoyo despite general poor sanitary conditions during processing procedures.

Surveys on consumption patterns showed that Munkoyo is mainly consumed as an energy drink by the entire population when hungry and at household level, whereas it is a social drink in urban areas. Processing practices and consumption patterns varied per sampling location. The fact that most rural communities consume Munkoyo as an energy drink when they are hungry and because it is easily prepared at household level with readily available raw material, make Munkoyo an ideal beverage to promote food security. This has also been observed for similar beverages like Kenkey, Mawe, Ogi and Akpan in West Africa [2, 26].

The profiling of bacterial communities revealed that the most dominant microbial species in this research include *Streptococcaceae*, *Leuconostocaceae*, *Enterobacteriaceae*, *Lactabacillales*,

**Table 4. Aroma compounds in Munkoyo from all surveyed locations with different commonly used inoculum treatments.**

| Aldehydes | Esters |
|---|---|
| 2,3-Butanedione | 1-Butanol,3—methyl-,acetate |
| 2,3-Pentanedione | 2,3-Butanedioldiacetate |
| 3-hydroxy-2-Butanone | Acetic acid, 2-phenylethyl ester |
| 2-n-Pentylfuran | Butanoic acid, ethyl ester |
| Hexanal | Ethyl Acetate |
| **Organic acids** | Nonanoic acid ethyl ester |
| 9-Hexadecenoic acid | Octanoic acid, ethyl,ester |
| Acetic acid | Propanoic acid, 2-hydroxy-ethyl ester |
| Butanoic acid | **Alkanes** |
| 3-methyl- Butanoic acid, | 4,6-Dimethyldodecane |
| n-Hexadecanoic acid | 9H-Flouren-9-yl(trimethyl) silane |
| Octadecanoic acid | Hexadecane |
| Oleic Acid | Hexane |
| Tetradecanoic acid | n-Tetradecane |
| **Alkenes** | **Alcohol** |
| (5Z)-2,6,10-trimethyl-1,5,9-undecatriene | Ethanol |
| 1-ethyl-2,4-dimethyl- Benzene | p-meth-1-en-8-ol |
| Cobalt,pentamethylcyclopentadiene | |
| Naphthalene | |

*Bacillaceae*, and *Aeromonadaceae*. The characterization of the microbial communities of the samples revealed high levels of diversity of bacteria within samples and high variation in bacterial community structure between samples. The Shannon index of up to 1.18 with an effective number of species of 3.44, indicates a relatively high diversity within bacterial communities. Like many other cereal-based fermented beverages, Munkoyo is largely spontaneously fermented, dominated by lactic acid bacteria [14]. Previous work on the microbial community structure of Munkoyo reported a similar bacterial composition [15]. Other studies evaluating the microbial communities of cereal-based fermentations reveal a great diversity of microbial communities with lactic acid bacteria being dominant [2]. This is in line with studies on cereal-based traditional fermented foods and gruels such as Ogi from Nigeria, Mawe and Akpan from Benin and Togwa from Tanzania [4, 7, 9, 11, 27].

Cluster analysis of factors that explain the variation in bacterial community structure in relation to processing variables and consumption patterns, revealed that how the product is consumed and sampling location determined consumption pattern and bacterial community structure, respectively. Processing variations of one hour cooking time, one day hydrolysis time and one day fermentation time exist, which apparently do not significantly alter microbial community composition. In previous work with only six Munkoyo samples [15], sampling location was not identified as a driver for differences between microbial communities, and the suggestion was made that processing practices could be important. In the present study, based on 96 bacterial community profiles, we find no evidence for processing practices as major driver of bacterial community structure, but rather that sampling location is most important.

The action of bacteria on fermentable sugars produces aroma compounds. Acetic acid was the most dominant aroma compound observed in all samples analysed. This could be due to the presence of acetic acid bacteria in biofilms in the fermentation vessels forming a microbial community with other microbes that produce acetic acid [28]. Other researches show that the *Acetobacteraceae* family is characterized by their ability to metabolize carbohydrates, thereby

releasing the corresponding products (aldehydes, ketones and organic acids) and oxidizing ethanol into acetic acid in aerobic conditions [28, 29]. Munkoyo is mostly fermented in buckets or calabashes, which can produce biofilms that can harbor acetic acid bacteria. The fact that fermentation is spontaneous in open air facilitates the oxidation of ethanol into acetic acid. *Acetobacteraceae* being among the family of bacteria identified and ethanol among the aroma compounds produced in Munkoyo suggest the possibility of a quick conversion of ethanol into acetic acid by *Acetobacteraceae*. These findings highlight the fact that spontaneous fermentation constitutes diverse microbial communities, which can possibly lead to variation in sensorial attributes of the beverage like flavour and taste. Other aroma compounds in the 15 samples from the different provinces under study include aldehydes, esters, alkenes, alkanes, organic acids and alcohols. This was not different from Gowe, a traditional malted fermented sorghum beverage from Benin, which contained groups of alcohols, aldehydes, organic acids, esters, hydrocarbons, furan and phenol as a result of spontaneous fermentation [10].

The micro-organisms found in Munkoyo and similar products are known to produce compounds with antimicrobial activities that act against some diarrhoeagenic bacteria [3, 5]. These antimicrobial effects have been confirmed by studies that lactic acid bacteria are effective in reducing constipation severity and improve bowel movement frequency in constipated but healthy people after consumption of fermented foods containing a specific *Lactobacillus casei* strain [30]–this could potentially also apply to the bacteria found in Munkoyo. Further research could formalize these general findings for Munkoyo to establish potential health benefits of Munkoyo. The research could also include the addition of microbial specific strains known to have probiotic properties, such as *Lactobacillus rhamnosus* [3]. In addition, this work could build on work on other traditional fermented foods where strains have been added that produce vitamins such as vitamin B12 and vitamin K, which would lead to bio-fortification of the raw materials used to produce Munkoyo (maize) [31].

Other further research to determine the link between specific microbial communities and the aroma profiles responsible for the sensory attributes in Munkoyo should be undertaken. A first step in this research would be the isolation of bacterial strains from Munkoyo to allow tests of functionality of (mixes of) these bacteria. Based on the fact that we detected wide variation in the microbial communities underlying Munkoyo fermentation, we expect that different combinations of different lactic acid bacteria would be able to generate the desired texture and aroma profiles. This future research may also include yeast and is essential to design appropriate mixes of micro-organisms for the production of starter cultures for commercial production of fermented foods that optimally meet the desired sensorial quality characteristics. Finally, while the low pH of the final Munkoyo product ensures general inhibition of most pathogenic bacteria, research into food safety aspects concerning the survival and proliferation of pathogens is needed to facilitate formalization of traditional Munkoyo processing.

## Supporting information

**S1 Questionnaire. Information on different methods of processing Munkoyo and consumption patterns.**
(DOCX)

## Acknowledgments

We would like to thank the Agriculture Camp extension officers (CEOs) from the Ministry of Agriculture for organizing the farmers for this research and the respondents who willing filled in the questionnaires and prepared Munkoyo samples for analysis. Technical staff from

University of Zambia for the support rendered during extraction of DNA from the samples in the UNZA Laboratory. Erik Meulenbroeks who guided SP through some important semantics of the pipeline during analysis of the DNA sequences.

## Author Contributions

**Conceptualization:** Sydney Phiri, Sijmen E. Schoustra, Eddy J. Smid, John Shindano, Anita Linnemann.

**Data curation:** Sydney Phiri, Sijmen E. Schoustra, Joost van den Heuvel.

**Formal analysis:** Sijmen E. Schoustra, Joost van den Heuvel.

**Funding acquisition:** Sijmen E. Schoustra.

**Methodology:** Sydney Phiri, Sijmen E. Schoustra, John Shindano, Anita Linnemann.

**Project administration:** Sijmen E. Schoustra, John Shindano.

**Writing – original draft:** Sydney Phiri, Sijmen E. Schoustra, Anita Linnemann.

**Writing – review & editing:** Sydney Phiri, Sijmen E. Schoustra, Joost van den Heuvel, Eddy J. Smid, John Shindano, Anita Linnemann.

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
