## [Decision Letter · Decision Letter 0]

22 Aug 2019

PONE-D-19-16074

Fermented cereal-based Munkoyo beverage: processing practices, microbial diversity and aroma compounds

PLOS ONE

Dear Dr. Schoustra,

Thank you for submitting your manuscript to PLOS ONE. After careful consideration, we feel that it has merit but does not fully meet PLOS ONE’s publication criteria as it currently stands. Therefore, we invite you to submit a revised version of the manuscript that addresses the points raised during the review process.

We would appreciate receiving your revised manuscript by Oct 06 2019 11:59PM. To enhance the reproducibility of your results, we recommend that if applicable you deposit your laboratory protocols in protocols.io, where a protocol can be assigned its own identifier (DOI) such that it can be cited independently in the future. For instructions see: http://journals.plos.org/plosone/s/submission-guidelines#loc-laboratory-protocols

We look forward to receiving your revised manuscript.

Kind regards,

Edy de Brito

Academic Editor

PLOS ONE

Additional Editor Comments:

Dear Authors,

As you can see the reviewers recommended minior revisions. Therefore, I ask you to check the comments and make a response to each item raised by the reviewers.

2. Thank you for including the following Ethics Statement with your submission:

'The University of Zambia Ethics Committee approved the field work of this study'.

At this time, we ask that you please revise this statement to include any associated approval number(s) and please also add this statement to the beginning of your Materials and Methods section.

Reviewers' comments:

Reviewer's Responses to Questions

**Comments to the Author**

1. Is the manuscript technically sound, and do the data support the conclusions?

Reviewer #1: Yes

Reviewer #2: Yes

2. Has the statistical analysis been performed appropriately and rigorously? 

Reviewer #1: Yes

Reviewer #2: Yes

3. Have the authors made all data underlying the findings in their manuscript fully available?

Reviewer #1: Yes

Reviewer #2: Yes

4. Is the manuscript presented in an intelligible fashion and written in standard English?

Reviewer #1: Yes

Reviewer #2: Yes

5. Review Comments to the Author

Reviewer #1: Paper is, in general well writenn and experimental parts comptently done. Few suggestions/critical advices are reported below:

-giving the spontaneus fermentation and the presence of microbial family harbouring pathogenic species, I'm wondering how important is safety and if pathogens (and their concentration) were identified.

-fermentation process is even driven by yeast and yeast/bacteria interaction may be responsable for some identified flavour. I'm wondering if authors investigate (the yeast presence) this aspect as well.

- I'm usually consider, as effective tools and when it is possible, to integrate the NGS approach with cultivable methods. Your criticisms is gratly appreciated.

Line 382-390. This part is bit speculative since no antimicrobial activity was analysed as well as no vitamins concentrations determinate. Reference is mostly related to cobalamin. An example of bacteria produce K vitamin?

Reviewer #2: The manuscript titled "Fermented cereal-based Munkoyo beverage: processing practices, microbial diversity and aroma compounds" is well written and scientifically sound. Hence can be accepted for publication.

6. PLOS authors have the option to publish the peer review history of their article (what does this mean?). If published, this will include your full peer review and any attached files.

Reviewer #1: No

Reviewer #2: No

---

## [Author Response · Author response to Decision Letter 0]

17 Sep 2019

REPLY TO REVIEWERS COMMENTS

We thank the Reviewer for her/his thoughtful comments that have helped us to further improve our manuscript. Detailed responses to the points made are below. 

2. Thank you for including the following Ethics Statement with your submission:

'The University of Zambia Ethics Committee approved the field work of this study'.

At this time, we ask that you please revise this statement to include any associated approval number(s) and please also add this statement to the beginning of your Materials and Methods section.

***REPLY: This information has been added. Our approval letter did not have a number, we have added the date of the letter and the issuing body. 

***REPLY: A copy of a questionnaire has been attached. This is the original questionnaire that was used in our study. The original language of the questionnaire is English. 

*** REPLY: we will deposit the data in an open access depository and will provide the DOI upon acceptance of the manuscript. 

Reviewers' comments:

Reviewer's Responses to Questions

Comments to the Author

Reviewer #1: Paper is, in general well written and experimental parts competently done. 

Few suggestions/critical advices are reported below:

-giving the spontaneous fermentation and the presence of microbial family harboring pathogenic species, I'm wondering how important is safety and if pathogens (and their concentration) were identified.

***REPLY: The Reviewer is correct that the bacterial families/genera we found in Munkoyo are very diverse and that some families also include pathogenic bacteria. We also found that the pH of the products consistently is around pH 4 or lower. This indicates that the bacteria that lower the pH of their environment are a dominant factor in the microbiota. At this low pH, the proliferation of pathogenic bacteria is suppressed. The question of presence and survival of pathogenic bacteria is important for the further development of Munkoyo towards formalization and upscaling and should be part of future research. We have added a remark to this effect to at the end of the discussion section (starting line 403). 

-fermentation process is even driven by yeast and yeast/bacteria interaction may be responsible for some identified flavour. I'm wondering if authors investigate (the yeast presence) this aspect as well.

***REPLY: It is true yeast could inevitably be involved in the production of identified flavors. We chose not to focus on yeast in the present work since previous work on Munkoyo had revealed that yeast was not present in most Munkoyo samples that had been analysed (Schoustra et al 2013, PLOS ONE). We agree with the Reviewer that including yeast would however be of interest and we would recommend doing this in any follow-up to our present work and have added a remark to the discussion section (line 401). 

- I'm usually consider, as an effective tool whenever it is possible, to integrate the NGS approach with cultivable methods. Your criticisms is greatly appreciated.

***REPLY: We agree with the Reviewer that the use of culture based methods would be useful in the study of Munkoyo; we feel that this is especially the case in a next step of the research, namely the testing of which mixes of bacteria can generate products with properties similar to Munkoyo that is produces using spontaneous fermentation. We did not use culture based techniques in the present study since the target in this research was to initially identify all possible bacteria in the beverage, for which the use of non-culture based techniques has the highest resolution. We have added the suggestion of using culture based methods to the discussion section (starting line 396).

Line 382-390. This part is a bit speculative since no antimicrobial activity was analyzed as well as no vitamins concentrations determinate. Reference is mostly related to cobalamin. An example of bacteria produce K vitamin?

***REPLY: The Reviewer makes a valid comment; it is true no antimicrobial activity was analyzed and no vitamin concentration was determined in our study. However, these effects have been found in other studies and could also apply to Munkoyo given the bacterial community composition that we found. In response to comment, we have added remarks to this part of text that the antimicrobial activity and vitamin K production have been demonstrated in other studies and that this may also apply to Munkoyo (line 387 and text starting line 391).

Reviewer #2: The manuscript titled "Fermented cereal-based Munkoyo beverage: processing practices, microbial diversity and aroma compounds" is well written and scientifically sound. Hence can be accepted for publication.

---

## [Editor Report · Decision Letter 1]

24 Sep 2019

Fermented cereal-based Munkoyo beverage: processing practices, microbial diversity and aroma compounds

PONE-D-19-16074R1

Dear Dr. Schoustra,

We are pleased to inform you that your manuscript has been judged scientifically suitable for publication and will be formally accepted for publication once it complies with all outstanding technical requirements.

With kind regards,

Edy de Brito

Academic Editor

PLOS ONE
---

## [Editor Report · Acceptance letter]

15 Oct 2019

PONE-D-19-16074R1 

Fermented cereal-based Munkoyo beverage: processing practices, microbial diversity and aroma compounds 

Dear Dr. Schoustra:

I am pleased to inform you that your manuscript has been deemed suitable for publication in PLOS ONE. Congratulations! Your manuscript is now with our production department. 

With kind regards,

on behalf of

Dr. Edy de Brito 

Academic Editor

PLOS ONE